# Genome-wide discovery and phenotyping of non-coding transcripts in *A. fumigatus* reveals lncRNAs with a role in antifungal drug sensitivity

Danielle Weaver[1,6], Tanda Qi [2,3,6], Harry Chown [3,4,6], Marcin Fraczek[2,3], Ressa Lebedinec[3], Lauren Dineen[3], Clara Valero[3], Norman van Rhijn [3], Takanori Furukawa [3,5], Michael Bromley [3] ✉, Daniela Delneri[2,3] ✉ & Paul Bowyer [3] ✉

Recent data suggests one fungus, *Aspergillus fumigatus*, causes more deaths annually than HIV or malaria combined. Coupled with rapid emergence of antifungal drug resistance, the limited range of effective treatments, and mortality rates of >50%, aspergillosis represents a major challenge in infectious diseases. Recent studies have identified long-noncoding RNAs (lncRNAs) involved in drug resistance and virulence in pathogenic yeasts such as *Candida* spp. However, there is very limited knowledge of lncRNAs in human pathogenic moulds, including *A. fumigatus*. Here we exploit transcriptomics data of *A. fumigatus* exposed to different environments to annotate transcripts mapping to 2388 genomic loci. After manual curation we generate a database of over 1000 lncRNAs. We observe that the lncRNAs display orchestrated transcriptional profiles upon drug treatment and many are proximal to genes involved in azole sensitivity. We knock out a set of intergenic lncRNAs and perform a large-scale phenotypic analysis to identify 60 lncRNA mutants displaying condition-dependent fitness changes with 35 mutants exhibiting a positive growth phenotype under azole stress. Overall, this study generates and experimentally validates an important resource that will enable the wider research community to increase understanding of the functional importance of lncRNAs in *A. fumigatus*, including their involvement in drug sensitivity.

The ascomycete fungus *Aspergillus fumigatus* is one of the most serious pathogens of humans[1,2]. Disease incidence is low relative to TB or malaria but mortality is typically > 50% and recent data[3] suggests that *A. fumigatus* causes > 2 million deaths annually which would exceed malaria (630,000) and HIV/AIDS (640,000) deaths combined[4,5]. In common with other pathogens several recent projects have elucidated first the genome sequence of type isolates and more recently sequences of collections from clinics and the environment.

[1]Division of Infection, Immunity & Respiratory Medicine, School of Biological Sciences, Faculty of Biology, Medicine and Health, University of Manchester, Manchester, UK. [2]Manchester Institute of Biotechnology, The University of Manchester, Manchester, UK. [3]Division of Evolution, Infection & Genomics, School of Biological Sciences, Faculty of Biology, Medicine and Health, The University of Manchester, Manchester, UK. [4]MRC Centre for Global Infectious Disease Analysis, School of Public Health, Imperial College London, London, UK. [5]Present address: School of Health and Life Sciences, Teesside University, Middlesbrough TS1 3BX, UK. [6]These authors contributed equally: Danielle Weaver, Tanda Qi, Harry Chown. ✉e-mail: mike.bromley@manchester.ac.uk; d.delneri@manchester.ac.uk; paul.bowyer@manchester.ac.uk

There is a very limited range of effective treatments for aspergillosis and resistance to existing compounds is increasing rapidly, therefore a common focus of functional genomic studies has been elucidation of drug resistance. To date the annotation of the *A. fumigatus* genome has only undergone incremental changes from the original genome sequence annotation in 2005[6,7]. Recently, we published a telomere to telomere contiguous genome sequence for this pathogen allowing improved assessment of annotation and assignment of transcripts[8].

The conception of eukaryotic transcriptomes changed with the discovery that most areas of the human and yeast genome are transcribed, with long non-coding RNAs (lncRNAs) being common features of eukaryotic genomes[9,10]. lncRNAs are frequently associated with disease and disease susceptibility phenotypes in humans[11].

lncRNAs have been systematically identified in several yeast genomes, notably *Saccharomyces cerevisiae*[12] and *Schizosaccharomyces pombe*[13]. In budding yeast, libraries of ncRNAs mutants were constructed and functional analysis identified them as major players involved in regulatory and fitness networks across different environmental conditions[14,15]. lncRNAs involved in drug resistance and virulence have also been identified in yeasts[16] but the high degree of difference between lncRNA sequences in different species means that no functional orthologues can be identified except in very closely related species. Therefore, knowledge of lncRNAs involved in drug resistance cannot be inferred easily from comparative genomics. A recent study integrated publicly available transcriptomic data to predict thousands of novel lncRNA from five pathogenic *Candida* species[17]. However, although lncRNAs have been catalogued in several filamentous fungal species, notably *Neurospora crassa*[18,19], similar analyses have yet to be performed on a human pathogenic mould. Thus, in *A. fumigatus* the likely roles of lncRNAs in antimicrobial drug resistance (AMR) and virulence have not been systematically studied.

Small ncRNAs and housekeeping RNAs have been described in *A. fumigatus* to date[20–22]. However, the lack of systematically annotated lncRNAs remains a significant knowledge gap and we are likely missing an important component of the functional genomics landscape. In this study, we have used transcriptomics data to annotate over 8000 transcribed features mapping to 2388 loci in the *A. fumigatus* genome and generated an annotated database of over 1000 manually curated lncRNAs that can be used in transcriptomics studies which fills this knowledge gap and have used functional genomics approaches to demonstrate that lncRNA are a significant part of the antifungal drug response in *A. fumigatus*. lncRNAs display orchestrated transcriptional profiles and for the first time, we show that deletion of candidate lncRNAs leads to changes in azole drug sensitivity in this pathogen. Due to the increasing emergence of AMR, it is crucial to further our understanding of lncRNA which may have the potential to function as novel genetic targets. This study has generated a resource that will enable the wider research community to increase systematic understanding of the functional importance of lncRNAs in *A. fumigatus*, including their involvement in drug sensitivity.

## Results

### Genome-wide identification of novel long noncoding RNA in *Aspergillus fumigatus*

We generated a comprehensive transcriptome for *A. fumigatus* using RNA expression profiles of *A. fumigatus* strain A1163 in response to six antifungal drugs (miltefosine (MILT), 5-fluorocytosine (5FC), dodine (DOD), simvastatin (SIM), terbinafine (TERB), hygromycin (HYG)) across four minimum inhibitory concentrations (MIC) (0.5, 1, 2 and 4 X MIC). To identify novel transcripts, the resulting RNA-seq reads underwent a custom-made bioinformatics pipeline (Fig. 1). Initial transcript assembly identified over 46,000 transcripts covering ~6,900 loci. Once annotated genes and other features were removed 11313 transcripts at 2963 loci remained. Loci were then screened to

remove regions with coding potential leaving 8192 transcripts. Due to many features (loci) being represented by multiple overlapping transcript isoforms, there were 2388 loci in total. These 2388 loci were tested against a pooled RNA-seq dataset (*n* = 24 experiments) obtained from NCBI and shown to be able to serve as mapping targets for RNA-seq with a threshold of > 0.20 reads per feature. However, not all these features could be mapped reproducibly ( > 5 reads per feature) using the drug response dataset above. Therefore, representative transcripts were chosen based on highest read counts. Further filtering was performed, including a manual reannotation, to confidently identify 1089 novel lncRNA (Fig. 1 and Supplementary Data 1). We found that almost all lncRNA's are characterised by either being antisense ( ~ 60%) or intergenic ( ~ 40%), with a single sense lncRNA being the result of transcriptional-readthrough from a tRNA (Fig. 2A). The lengths of lncRNA ranged from 235 bp to 8.2 kb, with a median length of 1.8 kb and 1 kb for antisense and intergenic lncRNA, respectively (Fig. 2B). The majority of lncRNA contain a single intron, although we find a maximum of ten (Fig. 2C). In untreated samples, expression levels of lncRNA were lower than protein coding genes (PCG's) and around 2% were considered highly expressed (Transcripts per million (TPM) > 100), this is coherent with previous studies in humans and fungi[17,23] (Fig. 2D). We also found no significant difference in rate of expression between lncRNA classes (Fig. 2D).

### Sequence conservation of lncRNA only occurs within the *A. fumigatus* species complex

To assess the validity and conservation of the novel lncRNA in *A. fumigatus*, two publicly available transcriptome datasets (covering four *A. fumigatus* strains distinct from the A1163 reference used here) were mapped to the lncRNA. The majority (84%) of lncRNA were transcribed in at least one dataset (Fig. 2E). Each *A. fumigatus* strain contained at least 60% of the newly identified lncRNA, suggesting considerable conservation within the species. We assessed the sequence-level conservation of lncRNA in the *Aspergillus* genus (Fig. 2F and Supplementary Data 2). Only intergenic lncRNA were assessed due to the generation of false-positive mapping of antisense lncRNA to their respective sense-genes. Here we identified intergenic homologues in the genomes of two alternative *A. fumigatus* strains, 599 were present in Af293 and 613 in GbtcF1. This provides further validation of nucleotide sequence identity of novel lncRNA between strains. In contrast, analysis of genomes from 125 non-*fumigatus* Aspergilli, comprising the entirety of reference *Aspergillus* genomes from NCBI, reveal little interspecies conservation of lncRNA (Fig. 2F). 146 intergenic lncRNA were found to have sequence homology in other fungal genomes. 12 *Aspergillus* species were tested for the presence of close homologues (90% identity over 90% of the lncRNA, Fig. 2F). In total, 15 lncRNAs were found to be conserved in at least three species, with one lncRNA (*AFUBlnc_4940.C*) found within 9 species (Fig. 2F). The number of loci present within these species closely match the phylogenetic relatedness towards *A. fumigatus*[24,25]. Previous phylogenetic analyses of this genus have revealed that *A. oerlinghausensis*, *A. fischerii* and *A. fumigattaffinis* have a genetic similarity to *A. fumigatus* in, respective, decreasing order.

To determine whether the intergenic regions were expressed in other species even in the absence of homology we mined publicly available transcriptome datasets from *A. niger* and *A. nidulans*. We observe that 18 of 20 lncRNA regions analysed that lie between syntenic genes in these species also have lncRNA transcripts using the definitions previously given in this paper.

### lncRNA expression is regulated in a coordinated manner by treatment with antifungal drugs

To assess whether lncRNAs exhibited a coordinated expression response to external stimuli (as seen for protein coding genes) we reconstructed the lncRNA transcriptome from existing RNA-seq data

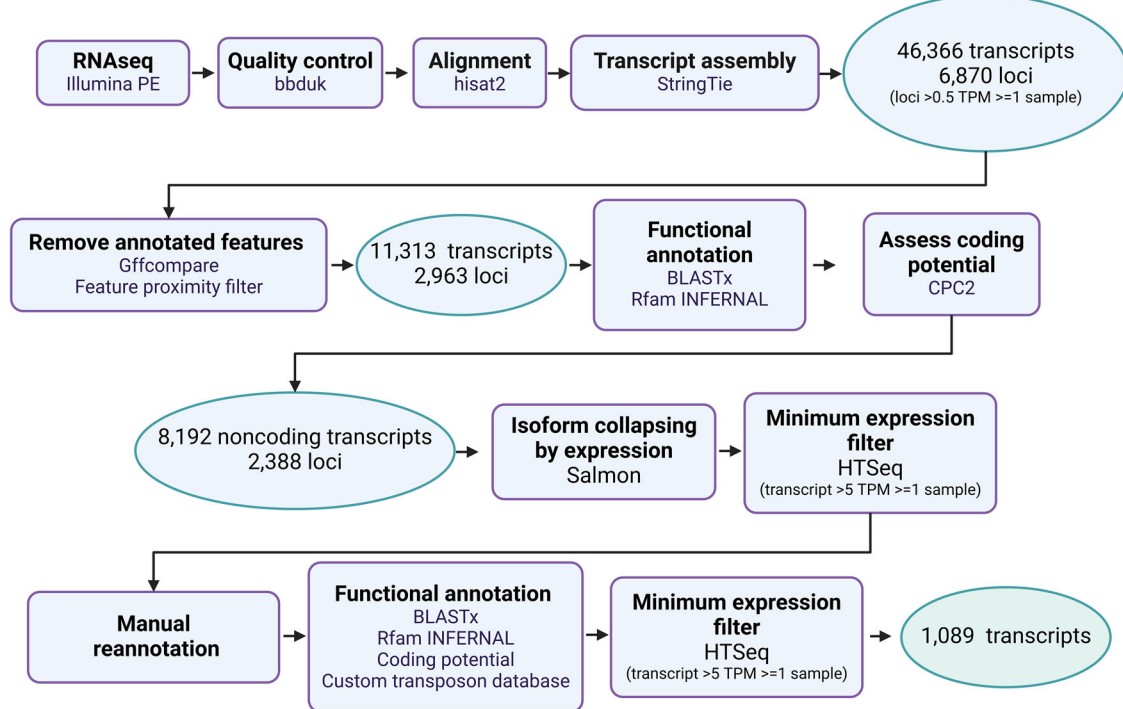

**Fig. 1 | Long noncoding RNA prediction pipeline.** Quality trimmed and filtered RNA-seq data were aligned to the *Aspergillus fumigatus* A1163 genome. Transcripts were assembled and subject to an expression filter (locus > 0.5 TPM in two replicates of at least one sample). Known annotated genomic features and those which immediately flank a gene (in sense) were removed. Noncoding transcripts were identified by removing features with sequence homology to known genes and ncRNA or those which have coding potential. The pipeline identified over 8000 novel noncoding transcripts. See Supplementary Fig. 1 for a summary of the output of the automated pipeline. The representative set was manually curated, and transcripts were re-annotated where necessary (functional annotation was repeated after this step). Finally, a transcript level expression transcript > 1 TPM in two replicates of at least one sample) filter was used to define the final putative lncRNA set, consisting of 1089 novel noncoding candidates. See Supplementary Data 1 for full annotation.

obtained from *A. fumigatus* with or without antifungal drug treatment. The datasets used include transcriptome responses from fungi exposed to sub- and super-MIC concentrations of azole antifungal. Transcript responses were quantified, and response profiles were clustered together indicating 15 different response profiles (Fig. 2G–I, and Supplementary Data 3, Supplementary note).

To determine whether patterns of expression of lncRNA are shared across the 7 different antifungal drugs, we clustered expression patterns in the sub- and super-MIC treatments for all antifungal drugs (Fig. 2J and Supplementary Data 4). The antifungal drugs tested cover a range of targets, including ergosterol biosynthesis (terbinafine [squalene epoxidase], simvastatin [HMG-CoA reductase], itraconazole [sterol 14α-demethylase], membrane disruption (dodine), lipid biosynthesis and intracellular transport (miltefosine), protein synthesis (hygromycin) and nucleic acid synthesis (5FC). Clustering identified several lncRNAs that show the same expression pattern in response to the drugs tested (Supplementary Data 5), with a single lncRNA, *AFUBlnc_7249.11*, exhibiting the same response across five drugs (miltefosine, itraconazole, dodine, simvastatin and hygromycin, see purple line in Fig. 2J). The drugs which shared the highest number of similarly expressed lncRNA were terbinafine and simvastatin (*n* = 66). Between 53–75% of expressed lncRNAs within each condition displayed a unique drug-specific expression profile, the most unique profile was found in hygromycin whereby 298 lncRNA had a distinct response towards the drug compared with others.

### lncRNA proximal to genes associated with azole sensitivity

To identify lncRNA with potential *cis*-regulatory roles in response to azoles, a set of genes known to be involved in azole sensitivity were assessed for their proximity to lncRNAs (Supplementary Data 6). Twenty genes were proximal to 32 lncRNA (*i.e.*, either inter-genic or

overlapping with CDS) including 18 genes with a neighbouring lncRNA(s) and 3 genes with an overlapping (antisense) lncRNA (Supplementary Data 7). The genes consisted of 9 ergosterol biosynthesis, 4 siderophore biosynthesis/transport, and 7 transcription factor genes. Six genes (including *cyp51A*) had two proximal lncRNA and two genes (*sidA* and *creA*) had three lncRNA. The identified lncRNA were mostly intergenic (28/31) and all but 4 intergenic lncRNA were upstream of the gene of interest. Of the 24 upstream intergenic lncRNA, 16 were bidirectional.

### Co-expression of proximal lncRNA and genes associated with azole sensitivity

Six genes (*creA*, *cyp51A*, *hapB*, *nctB*, *sidA* and *sidG*) were differentially expressed in response to itraconazole and had associated lncRNA(s) which were also differentially expressed (See Supplementary note). Cluster analysis found that three genes - *hapB*, *sidA* and *sidG* - had similar or opposing expression patterns to their corresponding lncRNA (Supplementary Data 7). The siderophore biosynthesis gene *sidA* is flanked by three lncRNAs (Fig. 3A), and all lncRNAs displayed co-expression (down-regulation) in response to itraconazole treatment (Fig. 3B). In addition, there was significant correlation between the log fold change (LFC) values of *sidA* and the nearby lncRNAs in all drug datasets excluding hygromycin (Fig.3C and Supplementary Fig. 5A). A siderophore biosynthesis gene cluster was found to contain a lncRNA (*AFUBlnc_3085.15*) which is antisense to both *sidG* and *estB* (Fig. 3D). The lncRNA and all the genes within the cluster displayed similar response patterns to both itraconazole and simvastatin (Fig. 3E). Furthermore, when combining the data from most drug treatments (excluding hygromycin) there was strong correlation between the LFC values of the lncRNA and each of the genes in the cluster (Fig. 3F).

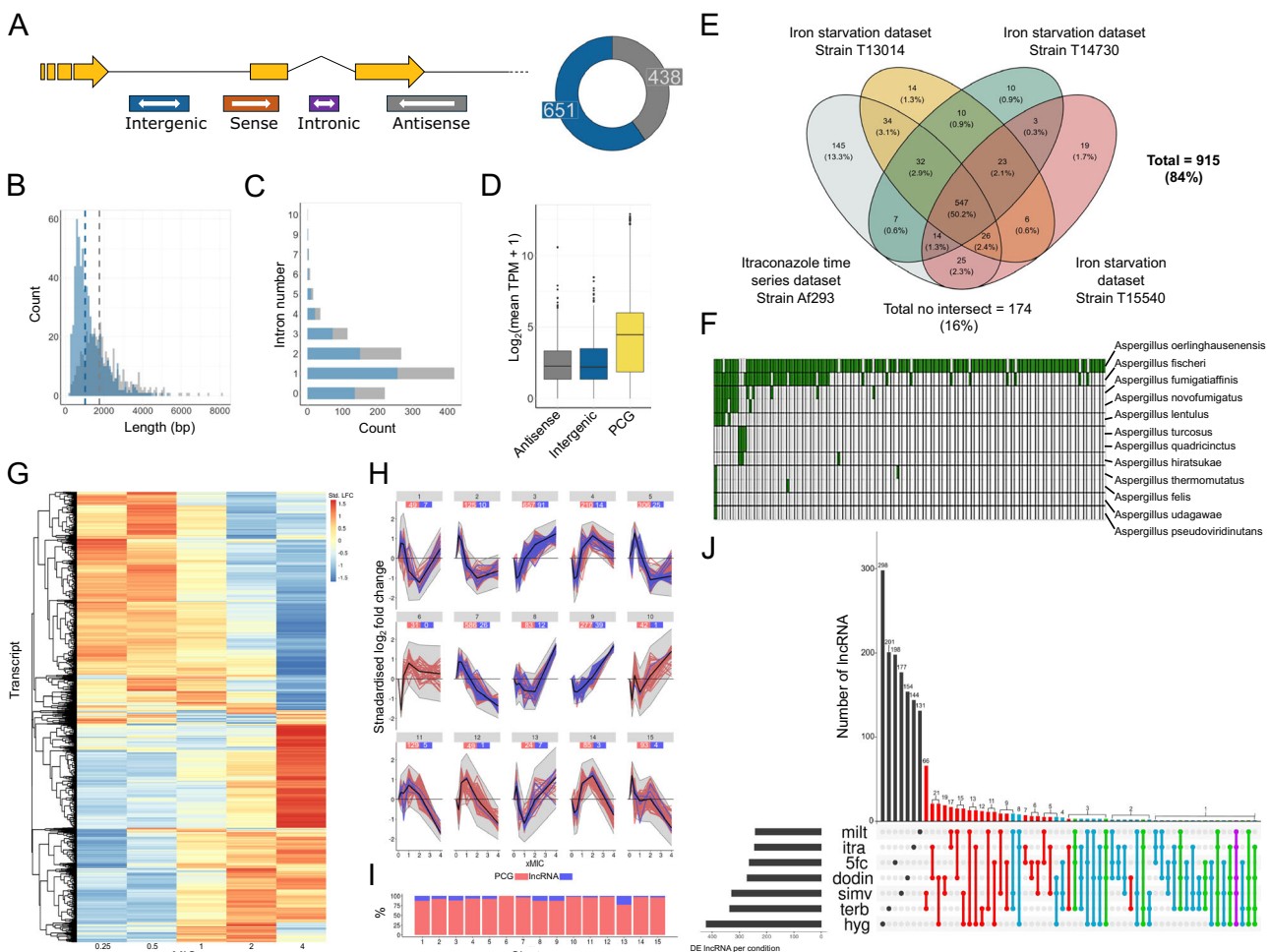

**Fig. 2 | Conservation and coordinated expression profiles of the *A. fumigatus* lncRNA. A** Classes of novel lncRNA and their genomic position, in respect to protein coding genes (PCGs). The pie-chart shows the distribution of these classes within our dataset, with the majority comprising of the intergenic (*n* = 651, 60%. Shown in blue) or antisense (*n* = 438, 40%. Shown in grey) class. **B** Nucleotide lengths of lncRNA. Median lengths for lncRNA, per class, are indicated by the dashed line (1.0 and 1.8 kb for intergenic and antisense, respectively). **C** Number of introns per lncRNA per class. **D** Expression levels of lncRNA (Grey; antisense, Blue; intergenic) and protein-coding genes (PCG, Yellow) in untreated samples. Expression levels expressed as log2(mean transcripts per million (TPM) plus 1).Boxplot centre lines indicate the median. The upper and lower hinges represent the upper quartile (Q3) and lower quartile (Q1), respectively. The upper and lower whiskers represent 1.5 times the interquartile range (IQR) above Q3 and below Q1, respectively. **E** Number of lncRNAs found in transcriptome datasets from two publicly available datasets, consisting of four *A. fumigatus* strains. **F** Binary presence-absence matrix showing the conservation of lncRNAs in non-*fumigatus Aspergilli*. Green indicates the presence of intergenic non-coding elements. **G** Heatmap of hierarchically clustered standardised log2-fold change (std. LFC) values of all differentially expressed transcripts across differing concentrations of itraconazole. Gradient colours represent degree of std. LFC from < −1.5, represented as dark blue to std. LFC > 1.5, dark red. **H** Line graphs from hierarchical *k*-means clustering of PCGs (pink) and lncRNAs (blue), with the median trend shown in black and the surrounding grey ribbon indicating ±1 standard deviation from the median. The total number of PCGs and lncRNAs are shown below each cluster title. **I** Distribution of PCGs and lncRNAs within each cluster as a percentage of transcripts within each cluster. **J** UpSet plot depicting the frequency of lncRNA that have the same patterns of expression upon treatment of individual antifungal conditions. Colours indicate the number of shared responses either one (black), two (red), three (blue), four (green) or five (purple) conditions.

Cyp51A is a sterol C14-demethylase and is a target of the azole antifungals. As a result, its mutation is a significant contributor to azole resistance. There are two lncRNAs, termed *cyp51A*-promoter lncRNA (CPL) and anti-CPL, found within the promoter region of *cyp51A* (Fig. 3G, H). Transcription from the lncRNA locus upstream of *cyp51A* was validated using qRT-PCR and showed both the lncRNA locus and *cyp51A* were transcribed at ~5-fold higher levels compared to the intergenic region that separates them (Fig. 3J). Clustering of gene and lncRNA responses upon itraconazole treatment did not identify correlated expression between either lncRNA and *cyp51A*. In the other drug datasets, it is apparent that CPL was consistently upregulated in response to drug treatment and shows distinct dose response expression patterns to *cyp51A* (Fig. 3I). In order to determine whether expression of the *cyp51A* upstream lncRNA had an effect of gene regulation the lncRNA region was cloned and expressed at a neutral trans locus. However, no impact on azole MIC was observed.

## lncRNAs have functional roles in resistance to azole drugs
To investigate the phenotypic importance of lncRNAs a subset were chosen to be deleted and their effects assessed under various stress conditions. Criteria for selection were TPM over 250 per lncRNA, > 200 nt away from flanking coding sequences and exhibiting > 2 log2-fold differential expression upon itraconazole treatment. This excluded the *cyp51A* and some other candidate azole relevant genes described above. Deletion mutants for 92 loci were generated and validated via spore PCR, and we were able to produce at least two biological replicates for 62 mutants. Large-scale phenotyping was performed via screening the relative fitness of each mutant compared to the

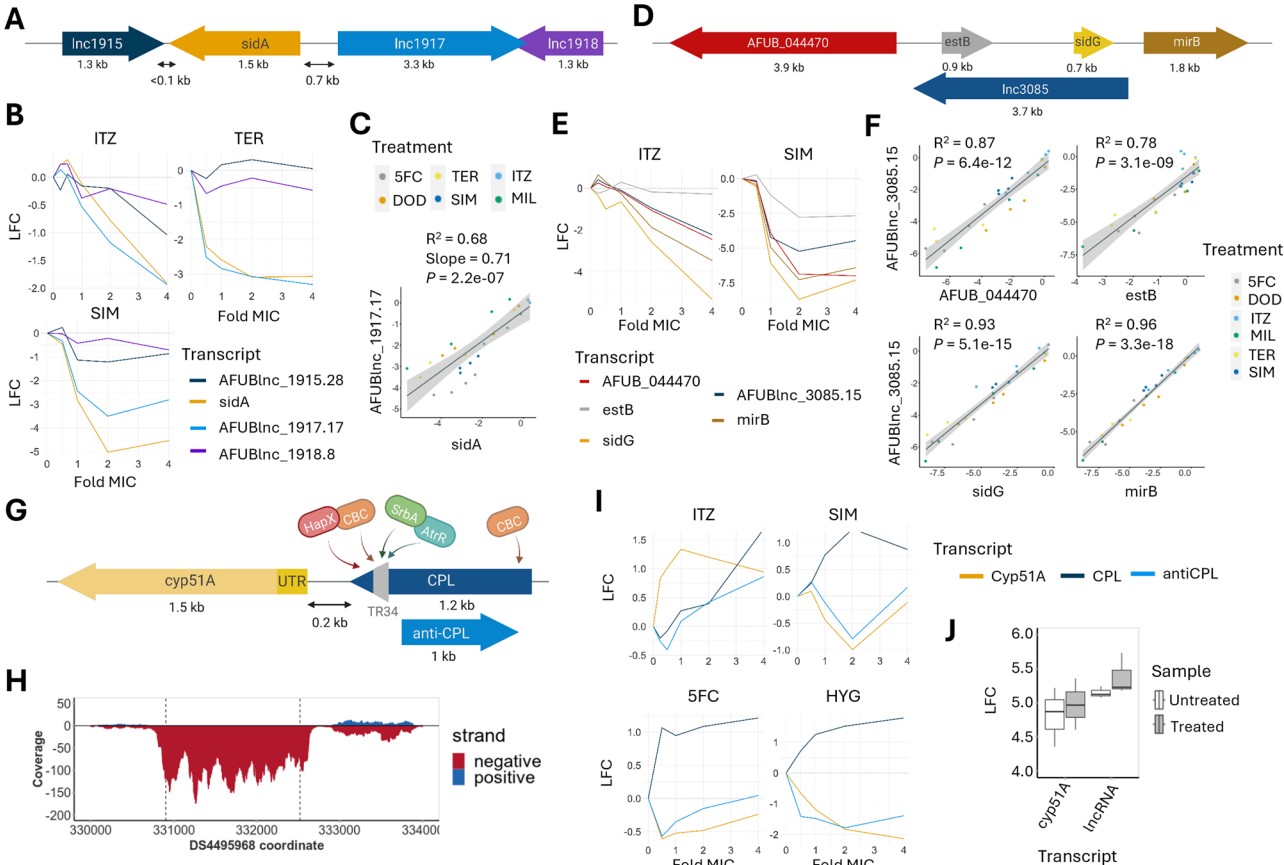

**Fig. 3 | LncRNAs are proximal to genes involved in azole-sensitivity and display correlated drug dose response expressions patterns. A** The L-ornithine N5-oxygenase gene, *sidA*, is flanked by one downstream lncRNA and two upstream lncRNAs. **B** Expression patterns of *sidA* and neighbouring lncRNAs in response to increasing concentrations of itraconazole, simvastatin and terbinafine. **C** Correlation between the log fold change (LFC) values of *sidA* and *AFUBlnc_1917.17* in response to varying concentrations (0.5,1-,2- and 4-fold MIC) of six antifungal drugs. Linear regression trend line shown in black with grey band indicating 95% confidence interval. **D** The fusarinine C acetyltransferase gene, *sidG*, is present in a cluster with other siderophore-associated genes *mirB*, *estB* and an ABC transporter, *AFUB_044470*. There is a lncRNA antisense to both *sidG* and *estB*. **E** Expression patterns of all transcripts within the *sidG* cluster in response to the antifungal drugs itraconazole and simvastatin. **F** Correlation between the LFC values of each nearby gene and the lncRNA *AFUBlnc_3085.15*. **G** Schematic representation of the two novel lncRNAs covering the promoter region of *cyp51A*, termed CPL and anti-CPL. CPL is 2

basepairs upstream of the *cyp51A* 5' untranslated region (UTR) and covers all known transcription factor binding sites of *cyp51A*. Linear regression trend line shown in black with grey band indicating 95% confidence interval. **H** Coverage plot showing an example of mapped RNA-seq reads in the region of *cyp51A*. Red indicates transcription from the negative strand (*cyp51A* and CPL) and blue indicates transcription from the positive strand (anti-CPL). **I** Expression patterns of *cyp51A* and its upstream lncRNAs in response to itraconazole simvastatin, 5-flurocytosine and hygromycin B. **J** qPCR validation of the increased expression of *cyp51A* and the lncRNA locus in comparison to the 200 bp intergenic region between them. Box-plot centre lines indicate the median. The upper and lower hinges represent the upper quartile (Q3) and lower quartile (Q1), respectively. The upper and lower whiskers represent 1.5 times the interquartile range (IQR) above Q3 and below Q1, respectively. ITZ itraconazole. TER terbinafine. SIM simvastatin. DOD dodine. 5FC 5-fluorocytosine. HYG Hygromycin B. MIL miltefosine. Schematics were created in BioRender. Weaver, D. (https://BioRender.com/ule96g1).

wildtype, A1160, strain. We were able to identify 60 lncRNA KOs with fitness changes that were condition-dependent (Supplementary Fig. 2, and Supplementary Data 8). Elevated temperature (45 °C), firm media (4% agar), and iron-starved conditions showed the least growth differences. Mutants with a growth change under oxidative stress (2 mM $H_2O_2$) predominantly exhibited impaired growth (Supplementary Fig. 2C). 35 lncRNAs exhibited a positive effect on growth under azole stress, with a single mutant Δ*AFUBlnc_2188.7* showing a fitness improvement across all azoles (Fig. 4A). Further validation of these results was performed by assessing radial growth of 11 mutants which showed higher than parental control growth in response to azoles. 9 out of the 11 strains grew faster in the presence of azoles (Fig. 4B), including Δ*AFUBlnc_2188.7* (Fig. 4C). As the deletion of lncRNA may influence nearby genes, we compared the expression levels of the *AFUBlnc_2188.7* flanking genes, namely *AFUB_031830* and *AFUB_031840*, in the WT and the Δ*AFUBlnc_2188.7* strains. This was carried out in the presence of the three tested azoles (*i.e.*, ITR, POS, and VOR) at the concentrations used in the high-throughput screening

(0.0625 mg/L of ITR, 0.025 mg/L of POS, and 0.25 mg/L of VOR). The Δ*AFUBlnc_2188.7* knockout showed no change in expression for either flanking gene under challenge with ITR or POS and no change in expression for *AFUB_038140* in the presence of VOR (Supplementary Fig. 6A). The *AFUB_038130* gene showed a ~1 $\log_2$ fold reduction in expression in the presence of VOR compared to the parental isolate however knockout of either flanking gene did not increase growth on solid media in the presence of any azole (Supplementary Fig. 6B). These results are also supported by RNA-seq data on WT in the response to itraconazole. The downstream flanking gene (*AFUB_031840*) is not significantly differentially expressed in response to any concentrations of itraconazole treatment, while the upstream flanking gene (*AFUB_031830*) is significantly downregulated in response to itraconazole only when exposed to a very high concentration (4X MIC; LFC -1.8, adjusted *p* value < 0.0001, Supplementary Data 3). It therefore seems unlikely that deletion of *AFUBlnc_2188.7* leads to the azole resistance phenotype via altering expression of the flanking genes.

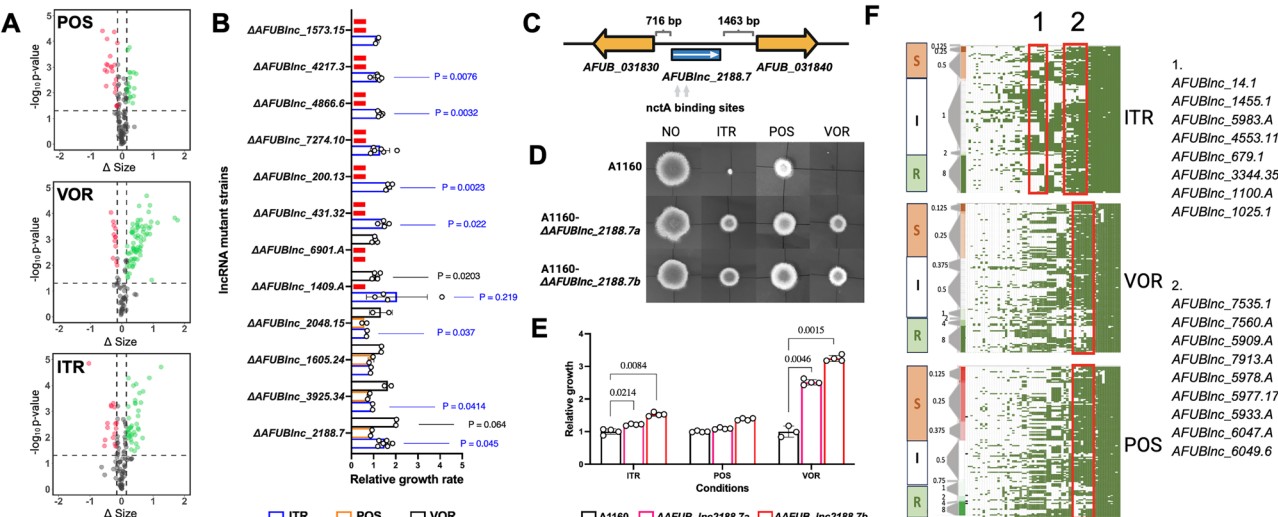

**Fig. 4 | LncRNAs are associated with azole resistance and show association with different levels of azole sensitivity on clinical strains. A** Relative growth changes of lncRNA knockouts are shown for media containing posaconazole (POS, 0.025 mg/L), voriconazole (VOR, 0.25 mg/L) and itraconazole (ITR, 0.0625 mg/L) in a high throughput screen. The x-axis represents normalised size change (Δ size) relative to standard condition; the y-axis represents the -$\log_{10}$ p value. LncRNAs with a significant change in growth ($p < 0.05$; $|\Delta$ size$| \geq 0.15$) are shown as red (fitness impaired) or green (fitness gain). P value was calculated using a two-tailed t-test. **B** Growth of selected strains from panel A on 9 cm petri-dishes. Radial growth is shown as fold change relative to the parental control and significant differences are shown. The red dashes represent ncRNAs not tested with the corresponding azoles. P values were calculated with paired t-test. Error bars represent SD. The number of biological replicates is indicated by the number of data points shown in each bar. Tests used 0.125 mg/L of ITR (blue), 0.025 mg/L of POS (orange) and 0.25 mg/L of VOR (grey). **C** Genomic location of A1160-*ΔAFUBlnc_2188.7*. **D** Growth of two independent transformants (a, b) of A1160-*ΔAFUBlnc_2188.7* under azole exposure is shown. **E** Radial growth rate relative to A1160 parental control for each azole. P values were calculated with paired t-test using four biological replicates. Error bars represent SD. **F** Presence/absence of the 418 lncRNA genes that were not present in all genomes. Rows show data from each genome arranged by row in order of drug MIC for each isolate for each of the three azoles. The same results are presented for ITR, POS and VOR but have been clustered by row data in a Euclidean manner prior to ranking by MIC. MICs were categorised according to clinical breakpoints into resistant, intermediate and sensitive and lncRNA gene presence or absence was assessed for each. MIC ranges are shown on the left of the diagram (mg/L). lncRNA homology is shown as green ( > 90% identity to the region) or white (< 10% identity). Genomes are presented as rows and lncRNAs as columns. Red boxes indicate lncRNA groups that display significantly altered abundance (multiple logistic regression, $p < 0.05$) in a comparison of resistant, intermediate and sensitive groups of isolates for ITR, VOR and POS. Source data are provided as a Source Data file. Schematics were created in BioRender. Qi, T. (https://BioRender.com/6ivohq6).

The lncRNA mutant *ΔAFUBlnc_2188.7* was investigated further; radial growth assays were performed on solid media (Fig. 4D, E). *ΔAFUBlnc_2188.7* showed a consistent increase in growth pattern in the presence of each azole compared to the wildtype, with the most notable difference in voriconazole (Fig. 4E). Although consistent and repeatable increases in growth were observed MICs using standard methods only reported modest increases in MIC, typically 2-fold.

**Presence of lncRNA genes corresponds to azole sensitivity**
To assess the possible role of lncRNAs in observed variations between azole sensitivity in *A. fumigatus* isolates, we compiled genomes from isolates (Supplementary Data 12) with known azole MIC for itraconazole, voriconazole and posaconazole and tested the presence or absence of lncRNA by homology (Fig. 4F).

We note that the lncRNA identified in the previous knockout experiment (Fig. 4A–E) as being involved in azole resistance are present in all genomes and therefore not included in this analysis. Presence or absence of the lncRNA genes in genomes of *A. fumigatus* was tested for association with each clinical breakpoint group, resistant, intermediate and sensitive, for each azole drug tested using Fisher's exact test adjusted for multiple sampling. 1051 of 1154 lncRNAs analysed were present in all genomes tested and 98 were present in <5% of genomes analysed leaving 56 genes that were tested for association with azole resistant phenotypes. Absence or presence of lncRNAs was significantly associated with drug resistance in each category. Patterns of resistance were initially visualised using hierarchical clustering suggesting two groups that might share resistance profiles. Group 1 and 2 are shown in Fig. 4F; absence of the lncRNA is indicated by lack of green colour and groups are outlined in red boxes for clarity. Although clustered not all lncRNA in each group show significant association with azole resistance when analysed individually with correction for multiple sampling (Supplementary Fig. 7).

## Discussion
The lncRNA database for *A. fumigatus* is a comprehensive collection of annotated lncRNAs covering the entire *A. fumigatus* genome. 2388 new lncRNA features were annotated, and we were able to assess transcriptional regulation and further refine the annotation for 1089 of these. The entire database is novel with only a small number of non-coding RNAs previously described in this organism. Excluding transfer and ribosomal RNAs, 45 small ncRNA transcripts ( < 500 nucleotides in length) were previously known in *A. fumigatus*. The majority (67%) of these small ncRNAs were small nuclear RNA (snRNA) and small nucleolar RNA (snoRNA), and these transcripts are not included in the lncRNA database described here[20]. The newly observed transcriptional landscape increases the proportion of the genome that is transcribed from 64% to 81%. The number of transcripts increased from 10,038 to a transcriptome of 12,350 features. Most lncRNAs identified were intergenic and overlapped coding genes in an antisense manner. We were unable to identify sense lncRNAs as fungal genes are compact with small introns which precludes discovery with existing protocols, although this does not exclude the existence of such transcripts.

Functionally our analysis found that only a small number of lncRNAs were conserved at the sequence level even in closely related species of the *Aspergillus* genus and that there was considerable variability for about 300 lncRNAs even within genomes from *A. fumigatus*. This suggests that lncRNAs should be included in any consideration of the *Aspergillus* pan-genome. We also noted that even in

cases where there was no sequence conservation between lncRNAs, when comparing syntenic areas of the genomes the loci were still transcribed, suggesting conserved function that requires active transcription of the area.

Transcription of the 1,089 lncRNAs that could be reliably statistically modelled in our antifungal drug response transcriptional datasets showed orchestration of transcription of 15 groups of lncRNAs comprising the complete dataset in response to different levels of azole, ranging from sub-inhibitory to strongly inhibitory concentrations. Modelling with additional drug datasets found expression patterns which were drug-specific, and also those which were shared in response to two or more drugs. Hygromycin is the only drug within the set which targets protein synthesis, and it displayed the highest number of uniquely responsive lncRNA. In contrast, simvastatin and terbinafine displayed the most lncRNA with shared responses. This could be explained by the fact that both drugs target the ergosterol biosynthesis pathway, and at earlier points in the pathway compared to azole drugs.

We carried out a large-scale phenotypic analysis of candidate intergenic lncRNAs. A set of 92 intergenic loci were knocked out and we identified 60 lncRNA mutants displaying condition-dependent fitness changes in response to drug with 35 mutants exhibiting a positive effect on growth under azole stress. The effects on radial growth were strong although there was limited change when resistance was assessed using a clinical MIC method. This concords with previous work in this fungus where overlapping and related mechanisms of drug sensitivity such as efflux transporters are knocked out[26]. In such cases changes on growth are clear but there is limited or no reproducible change in MIC. A further issue may be that removal of an intergenic lncRNA may interfere with the regulation of neighbouring genes and therefore that observed phenotypes are confounded by *cis* effects. Such issues have largely been addressed by the yeast and human lncRNA community. Here, the lncRNA knockout with the strongest effect on azole sensitivity (*i.e., AFUBlnc_2188.7*) overlaps with two known binding sites for a transcription factor which is known to affect azole sensitivity[27], although we note that such binding sites occur in several genes not involved in drug response. When assessing the neighbouring gene expression in *ΔAFUBlnc_2188.7* under azole exposure, we found only one condition-gene pair showing a small but significant expression change ( ~ 1 $\log_2$ fold reduction for *AFUB_013830* in the presence of voriconazole). Given the deletion of *AFUB_013830* does not affect azole sensitivity, the reduced expression observed is unlikely to contribute to the azole resistance trait. Thus, the azole resistance phenotype associated with deletion of *AFUBlnc_2188.7* is unlikely to result from altered expression of its flanking genes, but rather from its function as a lncRNA. It is possible that the function of the lncRNA is to maintain an open chromatin structure at this site through being transcribed. There is a wide variation in basal drug sensitivity of different strains of *A. fumigatus* often from environmental sources that do not carry known azole resistance mutations and that are regarded as drug sensitive or intermediate in clinical parlance. We observed that several lncRNAs were present or absent in groups of genomes that had been arranged according to their basal drug sensitivity and we suggest that this variability might arise from presence or absence of these genes. Such variability is thought to be important in clinical situations involving sub-optimal therapy. Proof of this conjecture will require careful further study in the context of the pan-genome of these isolates to avoid confounding factors.

The lncRNA genes will also form an important part of annotation of SNPs in GWAS studies as many previously reported SNPs lie within these regions. We provide accurate and updated annotation files that can be directly used in transcriptome and GWAS experiments to include function and transcription of non-coding features in future work.

## Methods

### Sample collection, library preparation and sequencing

Data input to the lncRNA prediction pipeline is from a multidrug in vitro experiment. *Aspergillus fumigatus* CEA10 spores were inoculated at $1 \times 10^6$/ml into 50 ml Vogel's minimal media with 1% glucose and incubated at 37 °C, 180 rpm for 16 h. Mycelia (1 g/culture) were then shifted to 50 ml of RPMi-2% glucose induction medium and incubated at 37 °C, 180 rpm for 4 h in the presence or absence of drug treatment (27 sample conditions in total). This dataset comprises three groups (each with an untreated control data triplicate) of samples treated with six antifungal drugs (simvastatin, miltefosine, terbinafine, hygromycin, 5-fluorocytosine and dodine) at four concentrations (0.5, 1, 2 and 4X minimum inhibitory concentration (MIC)). A second RNA-seq dataset from an itraconazole treatment experiment was used for differential expression analysis. This was performed as described previously, except the drug treatment was performed at five concentrations (0.25, 0.5, 1, 2 and 4X the MIC). RNA was prepared as previously described[27]. Sequencing libraries were prepared using Illumina TruSeq stranded mRNA kit and sequenced (2 × 100) on an Illumina HiSeq.

### lncRNA prediction pipeline

**Quality control & transcript assembly.** Bbduk[28] (14/08/18) was used to perform the following on all RNA-seq data: trim to Q20, filter to Q25, filter for minimum length of 50 and trim polyA tails of at least 5 adenines.

hisat2 (v2.1.0) was used to generate stranded alignments (maximum intron length of 5 kb) against the A1163 genome. Each alignment was used by Stringtie (v2.0) to generate both de novo and genome-guided transcriptome assemblies which were subsequently merged to create a final nonredundant assembly. Stringtie expression mode was used to calculate TPM values, and transcripts with a TPM of 0.5 or more in two replicates of a sample were retained within the assembly.

**Exclusion of annotated features.** To enable the removal of all annotated features in A1163 genome, a comprehensive annotation file was created to be used as reference. First, the following annotations were mapped to the A1163 genome using Gmap[29] (v2018-07-04): FungiDB-47 annotated transcripts, Ensembl ncRNA annotations from *A. fumigatus* strain Af293 and a custom-made database of known transposon sequences. The master annotation file was then created by merging all the annotations using Gffcompare[30] (v.0.11.2). Gffcompare was then used to assess the genomic location of stringtie-generated transcripts with respect to annotated features. Transcripts deemed intergenic, intronic or antisense to a gene were carried forward. In addition, to reduce the likelihood of transcripts which are a continuation of a known gene which has insufficiently annotated ends, we removed any transcripts within 20 nt of a gene transcribed in the same direction using BEDtools[31] intersect (v.2.26.0).

**Functional annotation.** Functional annotation was performed to remove transcripts likely to be known coding or noncoding features. FASTA sequences were extracted from Ensembl *A. fumigatus* A1163 genomic fasta using gffread[30] (v0.11.7). To identify annotated protein coding genes, sequences were subject to a BLASTx[32] (v2.4.0) search against all fungal sequences in the nonredundant nucleotide database. To remove known ncRNA in the assembly, sequences were subject to an Rfam database search using INFERNAL[33] (v1.1.2). Finally, to define those with no coding potential, Coding potential calculator 2[34] was used.

**Representative lncRNA transcript assembly.** To remove isoform complexity from the assembly and simplify further analyses, we chose the isoform with the highest summed read counts (calculated using

Salmon[35] v1.2.1) to represent each lncRNA locus. The final representative set of transcripts was subject to an expression filter to keep only features with a TPM > 5 in two replicates of a sample. The overall features of this representative assembly were assessed and visualised using custom shell and R scripts (Supplementary Fig. 1, scripts are available at https://github.com/Danweaver1/lncRNA-prediction).

**Generation of final transcript assembly.** The annotation of lncRNA transcripts in the representative assembly was visualised using IGV[36] (v2.3.98) and merged replicate transcriptomic data from untreated and drug treatment samples. The lncRNA annotations were removed if the putative transcript appeared to overlap in the same direction with transcription of a known feature or manually curated to improve the accuracy of the annotation. Following this manual re-annotation step, all transcripts were again subject to a functional annotation filter and TPM filter as described above. The final lncRNA annotation is available in Supplementary Data 1. The lncRNA FASTA sequences and corresponding annotation files are available at (https://github.com/Danweaver1/lncRNA-prediction).

**Validation of final transcript assembly.** To assess the validity of the novel lncRNAs, two publicly available transcriptome datasets were mapped to the lncRNA sequences to determine if they were detectable in external datasets and other *A. fumigatus* strains. These datasets were a time series experiment of Itraconazole exposure in three *A. fumigatus* clinical isolates (PRJNA482512) and an iron starvation experiment in *A. fumigatus* CEA17 (PRJNA381768). Quality control and mapping of transcriptome data were performed as described above. TPM data were obtained by converting count data generated by htseq-count[37] with intersection set to strict (v.0.11.2). Transcripts were considered expressed in the dataset if they had a TPM of 5 or more in at least two replicates of a sample.

**Conservation of lncRNAs within the Aspergillus genus.** Conservation of intergenic lncRNA sequences was assessed by generating a database of reference *Aspergilli* genomes and querying the nucleotide sequences of the intergenic lncRNAs against the database. This was achieved by obtaining the reference genomes of 127 *Aspergilli* from NCBI (date accessed: 27th May 2022) using the NCBI Datasets tool (version 13.21.0) (https://github.com/ncbi/datasets), with the taxon "*aspergillus*" and –reference flags. A BLAST database was generated from these genomes and used to map intergenic lncRNA nucleotide sequences using nucleotide BLAST[32] (version 2.9.0), percentage identity > 90% and query coverage > 90% was then used to filter results.

**Strains and culture conditions**
*A. fumigatus* strain A1160 was used in this study. *A. fumigatus* strains were propagated on Sabouraud dextrose agar (SAB agar, Sigma-Aldrich, Gillingham, UK) in cell culture flash at 37 °C for 48–72 h. Spores were harvested by using sterile PBS-0.05% Tween 20 solution (PBST) and collected by filtering through 4 layers of Whatman filter paper to remove hyphal fragments. The density of spores was counted using a hemocytometer (Hawksley, UK). Spores were stored in PBST + 20% glycerol solution at -80 °C. *Aspergillus* minimum medium (AMM)[38] and RPMI 1640 (Sigma-Aldrich, Gillingham, UK) were used for the phenotypic experiments.

**Gene knockouts of lncRNAs**
Genomic deletions were achieved according to published protocols[39], and original lncRNA loci were replaced by the hygromycin B resistance marker cassette[40] by homologous recombination promoted by long flanking regions of about 1.2 kb (Supplementary Fig. 3). The marker cassette was then transformed into *A. fumigatus* strain A1160, transformants were selected on hygromycin B plates and validation

spore PCR was performed according to published protocols[41]. The mutants are labelled with their AFUBlnc identifier followed by B1, B2 or B3 to indicate the biological replicates. The mutants and the genomic positions of the lncRNA and knockout cassettes are presented in Supplementary Data 9. Spores of validated lncRNA KOs were stored at -80 °C.

**Large-scale phenotypic screening**
The large-scale phenotyping of the lncRNA mutants was assessed in solid media under the standard condition (AMM at 37 °C) and different stress conditions, including lack of iron (AMM without containing iron), oxidative stress (AMM containing 2 mM $H_2O_2$), high temperature (AMM, incubated at 45 °C), firm media (AMM containing 4% Agar) and antifungal stresses (AMM containing 0.0625 mg/L Itraconazole, AMM containing 0.025 mg/L Posaconazole and AMM containing 0.25 mg/L Voriconazole). A1160 spores and spores for each lncRNA mutant were harvested with PBST containing 20% glycerol, diluted to an $OD_{600}$ of 0.12 and 200 μL transferred into 96-well plates (4 plates in total) as the cell source plate. The cells were then transferred onto culture medium in PlusPlate (Singer Instruments Ltd.) using the RoToR HDA robotic pin system with three technical replicates. To avoid plate edge effects, colonies on the perimeter were excluded from analysis. The plates were incubated at the intended temperatures for 48 h before scoring the colony sizes. The relative fitness was calculated by normalising the median colony size of the three technical KO replicates to the A1160 colony size in the specific condition tested. The pipeline for the phenotypic analysis is summarised in Supplementary Fig. 4. Volcano plots were generated using ggplot2[42] based on the relative fitness change between each stress condition and standard condition, and the $p$ value was calculated.

**Phenotypic growth assays on solid media**
LncRNA mutant strains were spotted on Petri dishes containing AMM supplemented with different azoles. The concentration of each azole used in the petri dish growth assay was optimised to allow suboptimal growth of A1160. The following concentrations were used for itraconazole, 0.125 mg/L; posaconazole, 0.025 mg/L; and voriconazole, 0.25 mg/L. A petri dish was divided into 4 parts evenly by marking a cross at the bottom, and -1000 spores were spotted in the centre of the cross. A minimum of two biological replicates were performed for each KO mutant, alongside the technical replicates. The growth was referred to as the colony size, calculated by taking the average radius in four directions on the cross, recorded daily. Radial growth rates were calculated and statistical analysis performed as previously described[43]. For radial growth measurements of *A. fumigatus* strains deleted in *AFUBlnc_2188.7* neighbouring genes, plates were inoculated with $10^4$ spores per strain and allowed to grow for 4 days on solid AMM or AMM supplemented azoles (voriconazole, 0.25 mg/L; itraconazole, 0.0625 mg/L; posaconazole; 0.025 mg/L). Radial growth measurements were performed every 24 h. The growth of each mutant in each condition was compared with their respective growth in standard condition AMM.

**Minimum inhibitory concentration (MIC) assays and half maximal inhibitory concentration (IC50) measurement**
The MICs were determined by following the instructions outlined in EUCAST E.Def. 9.3 using the broth microdilution technique[44]. A standardised quantity of $5 \times 10^4$ spores was inoculated in the 96 well plates containing a gradient of drug from 16 mg/L to 0.03125 mg/L. The plates were read both visually and under the microscope after being incubated at 37 °C for 48 h to obtain the minimum inhibitory concentrations. $OD_{600}$ were taken for each plate on the same date. These measurements were used to calculate survival curves and determine the IC50 value[43].

## RNA extraction and qRT-PCR

A1160 and *ΔAFUBlnc_2188.7* strains were grown in *Aspergillus* minimal medium (AMM), with or without azole supplementation. A total of $5 \times 10^7$ spores were inoculated into 50 mL of medium. Mycelia were harvested after 48 h of growth and snap-frozen in liquid nitrogen.

Total RNA was extracted using a bead-beating method with TRIzol reagent (Sigma-Aldrich, UK), followed by RQ1 RNase-Free DNase treatment (Promega, USA) and RNA purification using the RNeasy Mini Kit (QIAGEN, Germany), according to the manufacturers' protocols. Complementary DNA (cDNA) was synthesised from total RNA using GoScript™ Reverse Transcriptase (Promega, USA) with oligo(dT) primers. Concentration and purity of total RNA and cDNAs were assessed using a NanoDrop Lite spectrophotometer (Thermo Scientific, USA). qPCRs were performed in a 10 μL final volume containing 10 ng of cDNA, 3 pmol of each primer, and 5 μL of iTaq™ Universal SYBR® Green Supermix (2X). Reactions were run on a Roche LightCycler® real-time PCR system with the following cycling conditions: 40 cycles of 15 seconds at 95 °C, 30 seconds at 60 °C, and 30 seconds at 72 °C. Each experiment included three biological replicates and three technical replicates per sample. All qRT-PCR runs included a no-template control (NTC) and a control lacking reverse transcriptase (-RT). Relative gene expression was calculated using Cq values normalised to the expression of housekeeping gene actin (*AFUB_093550*).

## lncRNA and gene expression analyses

**Differential expression analysis.** Alignments of transcriptomic data were generated as described above. Count data was generated using htseq-count[37], with intersection set to strict (v.0.11.2). Subsequent bioinformatics analyses was performed using R[45] (version 4.1.2) and line graphs have been produced using ggplot2[42] (version 1.2.0). Differential expression analysis was performed on all transcriptomic data generated from drug exposure experiments using DESeq2[46] (version 1.34.0). Transcripts were classed as differentially expressed if they met the following parameters, in at least one condition: (basemean > 30, adjusted $p$ value < 0.05, squared-$\log_2$fold-change > 1) | basemean > 5000. $\log_2$fold-change values obtained from differentially expressed transcripts in the itraconazole exposure experiments underwent standardisation prior to hierarchical iterative $k$-means clustering, $k = 20$, iterations = 100, bootstrap = 1000, Euclidean distance = 1. This process was then repeated after inversing $\log_2$fold-change data of lncRNAs. Pheatmap[47] (version 1.0.12) was used for hierarchical clustering and to produce heatmaps.

**Patterns of expression at lncRNA loci.** Using BEDTools[31] (version 2.30.0) we examined the surrounding genomic regions of the identified lncRNAs. First, we used the 'intersect' tool to uncover any antisense lncRNAs that may have multiple sense partners. Then we used the 'closest' tool to identify lncRNAs that are neighbouring protein-coding genes, within 5 kb. We then annotated any resulting protein-coding genes using the *A. fumigatus* A1163 Fungi Ensembl database[48] (release 53) via the biomaRt[49,50] package (version 2.50.3). The resulting lncRNA and neighbouring gene partners are listed in Supplementary Data 10, and the antisense lncRNA and sense gene partners listed in Supplementary Data 11. Additionally, transcription factor binding sites identified in several studies[27,51–53] were combined and their distances from novel lncRNAs were assessed through a custom bioinformatics pipeline incorporating BEDTools 'closest', binding sites were labelled as upstream if they fell within 200 bp of the lncRNA locus. Differentially expressed loci nearby protein-coding genes underwent enrichment analysis using FungiFun2[54].

**Expression patterns across multiple drug conditions.** The appropriate drug condition tags were used to label all differentially expressed lncRNA data. From this we then performed the same hierarchical $k$-means clustering used on the itraconazole dataset. Then we used the UpSet[55] (version 1.4.0) and venn[56] (version 1.10) packages to identify the number of lncRNAs which had shared expression profiles across multiple drug conditions.

**lncRNAs proximal to azole-associated genes search.** To identify lncRNAs which are proximal to azole-associated genes, the closest protein coding neighbour of a lncRNA was first identified (upstream and downstream, within 10 kb distance limit) using the BEDtools[31] closest tool. The resulting list of lncRNA:gene pairs were searched to identify any azole-associated genes using the gene IDs in Supplementary Data 6. Expression data for the transcripts was visualised as described above.

## Reporting summary

Further information on research design is available in the Nature Portfolio Reporting Summary linked to this article.

## Data availability

Raw sequence data has been deposited at the NCBI sequence read archive (SRA) under accession number PRJNA861909 (ncbi.nlm.nih.gov/bioproject/861909) and PRJNA1165181 (ncbi.nlm.nih.gov/bioproject/1165181). Source data are provided with this paper, either in the supplementary Data or the Source Data File. Source data are provided with this paper.

## Code availability

Code used for analysis is available at https://github.com/Danweaver1/lncRNA-prediction (https://doi.org/10.5281/zenodo.17477034) and https://github.com/harrychown/asp_lncrna (https://doi.org/10.5281/zenodo.17416843) and https://github.com/Tanda-Qi/Analysis_Phenobooth_output (https://doi.org/10.5281/zenodo.17900633).

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

## Acknowledgements

D.W. is funded by the NIHR Manchester Biomedical Research Centre (BRC) (NIHR203308). This work was funded by the Wellcome Trust under project number 208396/Z/17/Z (to M.J.B., P.B. and D.D.).

## Author contributions

P.B., D.D., M.B. conceived and planned the experimental work, P.B., D.D., T.Q., D.W., H.C., C.V. and M.B. wrote the manuscript, D.W., T.Q., M.F., C.V. and T.F. performed laboratory work, D.W., H.C., P.B., T.Q., C.V. analysed the data and D.W., H.C., R.L., L.D. and Nv.R. performed additional manual annotation steps.

## Competing interests

The authors declare no competing interests.
