## [Transparent Peer Review file · Nature Communications]

Genome-wide discovery and phenotyping of non-coding transcripts in *A. fumigatus* reveals lncRNAs with a role in antifungal drug sensitivity.

Corresponding Author: Professor Paul Bowyer

Version 0:

Reviewer comments:

Reviewer #1

(Remarks to the Author)

In this manuscript, the authors generate a first-of-its-kind dataset annotating over 1000 new long noncoding RNAs in the fungal pathogen, *Aspergillus fumigatus*. Additional analyses uncover that the majority of these lncRNAs are only conserved at the species level and display drug-responsive orchestrated transcriptional profiles. The authors also report the deletion and characterization of all intergenic lncRNAs to identify 60 involved in condition-dependent fitness. Importantly, 11 of these lncRNAs displayed fitness changes in response to triazole antifungals, suggesting novel roles for these newly defined genetic elements in drug susceptibility. Although the fitness changes in response to antifungals is quite modest among the individual mutants analyzed here and is limited to non-clinical methods of detection, the novelty and expected impact of this dataset is high. The authors' implication that multiple lncRNA-mediated mechanisms may work in a concerted fashion to generate heterogeneity of drug susceptibility among clinical and environmental isolates is well stated. The studies are overall well-done, and the manuscript written in a concise manner given the large dataset. I have only a few minor comments for the authors to consider:

1. At line 104, I think the authors mean to say 0.5, 1, 2 and 4 "X MIC". Not just "MIC". Correct?
2. At lines 212-222, there is a lot of presentation concerning the drug conditions under which there were the most vs. least changes in expression profiles, etc. However, there is then no text in the Discussion that speculates on what any of this may mean with respect to what those drugs might be doing to the cell, etc. It would be helpful if the authors could expand this section a bit or add to the Discussion regarding this.
3. At lines 402-404 in the Discussion, it is not clear why redundancy in gene function would impact drug susceptibility when measured by broth microdilution vs. by radial growth on solid agar. Unless this point can be expanded for clarity, it does not really add to the data interpretation and could be removed.

(Remarks on code availability)

Reviewer #2

(Remarks to the Author)

In the study "Genome-wide discovery and phenotyping of non-coding transcript in *A. fumigatus* reveals lncRNAs with a role in antifungal drug sensitivity", the authors use a novel bioinformatic pipeline to rigorously annotate lncRNA genes in *Aspergillus fumigatus*. Analysis of RNA seq data revealed that lncRNA gene expression resolves into clusters, with antifungal drug treatment coordinating regulation of a large number of lncRNAs. The authors identify lncRNA loci that are proximal to azole resistance genes, focusing on two lncRNA genes that overlap with the *Cyp51A* locus promoter region. Expression of those lncRNAs in trans did not alter azole susceptibility. The study ends with construction of a deletion collection and assessment of those deletions on azole sensitivity. Follow up phenotyping of a deletion in *Inc2188.7* did reveal increased radial growth in the presence of azoles, but no significant change in resistance breakpoints by broth microdilution. The analysis that follows looking at presence/absence across strains is not super clear as presented. Overall,

the study offers a comprehensive annotation of lncRNA genes in *A. fumigatus*, and provides a deletion collection resource that will be useful for investigation of lncRNA function in the future. The outcomes presented in this paper fall short of increasing our understanding of lncRNA biology in this pathogenic mould.

1. The authors use the “co-localize” to indicate that lncRNA-encoding loci proximal to gene loci involved in azole sensitivity. To me, co-localization of a lncRNA implies where the transcript is, not the locus. I would suggest being specific about the chromosomal location of lncRNA genes, vs. just co-localization of lncRNAs.

a. The authors use lncRNA and lncRNAs as plural. lncRNAs is more common as the plural.

2. In line 249, CPL is used without definition. It is not defined except in the Figure 3 legend. Please define in the text.

3. The data contained in figure 3 are mostly not reported in the text, but rather rely on the figure legend to make conclusions rather than speak to the data being presented. The results in Figure 3 should be presented in the text and the legend revised to speak to the data presented in the figure. This is especially true for 3I.

4. The reporting of the results for Figure 4F is terse. For clarity, each group should be presented alone with its respective conclusion and directionality. What does “significant difference in presence or absence” mean?

5. In extended figure 4, there appears to be significant geographic effects on phenotype, with colonies on the perimeter of the plate exhibiting larger overall area than those in the internal spaces of the plate. Was this accounted for in the analysis?

(Remarks on code availability)

Code is available on github. I'm not a bioinformaticist, so am unable to review the code.

Reviewer #3

(Remarks to the Author)

This manuscript describes a well done, large-scale discovery of lncRNAs associated with antifungal exposure in *Aspergillus fumigatus*. This manuscript represents an enormous amount of work that has the potential to serve as a fantastic resource for the *Aspergillus* community. Overall, the manuscript is well written and clear.

One major aspect of this work is that the investigators generated deletion mutants for 92 of the lncRNA's and measured the growth of each mutant under several conditions. However, the authors do not go far enough to show that the observed phenotypes are the result of deleting a lncRNA. The best way to do this would be to generate a complemented strain where the lncRNA is expressed from another locus, as is done when analyzed protein-coding genes. I acknowledge that this may not always work and would depend on the exact mechanism by which the lncRNA is carrying out its function (which at this point is unknown). However, the investigators would at least need to demonstrate that deletion of a lncRNA does not alter the expression of the flanking neighboring genes. This is especially important given the authors observation that lncRNAs are co-localized with genes involved in azole-sensitivity. Systematically performing qRT-PCR on each of the flanking genes in each mutant, under the appropriate conditions dictated by the phenotype, would go along way towards addressing this concern. For example, if the expression levels of AFUB_031830 and AFUB_031840 were the same when comparing the Wt strain and the AFUBInc_2188.7 deletion mutants grown in the presence of ITR and VOR, one can be convinced that the increased growth is not simply the result of altering the expression of one of this protein-coding genes. The authors begin to address this by citing unpublished results obtained with deletion of AFUB_031830, but fall short. The finding that expression of AFUB_031840 is not altered by azoles is not proof that it is not involved. As currently presented, the assertion that “lncRNAs have functional roles in resistance to azole drugs” is not substantiated by the data.

(Remarks on code availability)

Version 1:

Reviewer comments:

Reviewer #2

(Remarks to the Author)

The authors adequately addressed the concerns of this reviewer.

(Remarks on code availability)

Reviewer #3

(Remarks to the Author)

The authors have adequately addressed my comments. This is an excellent manuscript. Very well done!

(Remarks on code availability)

Answers to Reviewers

The authors wish to thank the **Reviewers** and the **Editor** for taking their time to review our work and for providing comments and suggestions to improve the manuscript. It has really helped cement the core message of the paper and support the findings. Briefly we have addressed the reviewers comments below and performed additional experiments as suggested by the reviewers in order to address their points, also outlined below and presented in the rewritten paper.

Reviewer 1 (R1)

R1 has a few minor comments for us to consider:

Q1. At line 104, I think the authors mean to say 0.5, 1, 2 and 4 “X MIC”. Not just “MIC”. Correct?

A1: Yes, the R1 is correct. This has now been corrected (p4, line 107)

Q2. At lines 212-222, there is a lot of presentation concerning the drug conditions under which there were the most vs. least changes in expression profiles, etc. However, there is then no text in the Discussion that speculates on what any of this may mean with respect to what those drugs might be doing to the cell, etc. It would be helpful if the authors could expand this section a bit or add to the Discussion regarding this.

A2: The authors thank the R1 for pointing this out. A summary of the different aspects targeted by the drugs has now been added to the results section (p6, lines 173-176), and a description added to the discussion (p11, lines 326-332).

Q3. At lines 402-404 in the Discussion, it is not clear why redundancy in gene function would impact drug susceptibility when measured by broth microdilution vs. by radial growth on solid agar. Unless this point can be expanded for clarity, it does not really add to the data interpretation and could be removed.

A3: We agree with R1 and have removed this specific discussion point.

Reviewer 2 (R2)

Q1. The authors use the “co-localize” to indicate that lncRNA-encoding loci proximal to gene loci involved in azole sensitivity. To me, co-localization of a lncRNA implies where the transcript is, not the locus. I would suggest being specific about the chromosomal location of lncRNA genes, vs. just co-localization of lncRNAs.

The authors use lncRNA and lncRNAs as plural. lncRNAs is more common as the plural.

A1: The terms “co-localise/co-localised with” have now been changed to “proximal/ proximal to” throughout the manuscript; “lncRNAs” has now been included as plural where appropriate.

Q2. In line 249, CPL is used without definition. It is not defined except in the Figure 3 legend. Please define in the text.

A2: The definition of CPL has now been added to the text (p8, line 217).

Q3. The data contained in figure 3 are mostly not reported in the text, but rather rely on the figure legend to make conclusions rather than speak to the data being presented. The results in Figure 3 should be presented in the text and the legend revised to speak to the data presented in the figure. This is especially true for 3I.

A3: A more detailed description of the data for figure 3 was originally included in supplementary text, but after the **R2**'s comment, we realise that the supplementary text was not cross-cited this in the main text.

The figure legend has now been revised to describe the data presented (p26-27), and the main conclusions added to the results text (p7-8). Further explanation remains within the supplementary text, and this is now cited in the results text (p7, line 202).

Q4. The reporting of the results for Figure 4F is terse. For clarity, each group should be presented alone with its respective conclusion and directionality. What does "significant difference in presence or absence" mean?

A4: The reviewer is correct to highlight this point. Text in the main body of the paper has been rewritten for clarity. One supplementary figure and one supplementary table have also been added - one (Extended Data Figure 7) to show the detail of the statistical association for each lncRNA in the analysis with resistance to the three azole drugs and the second table (Supplementary Table 12) to show the origins of the genomes used in the analysis.

Q5. In extended figure 4, there appears to be significant geographic effects on phenotype, with colonies on the perimeter of the plate exhibiting larger overall area than those in the internal spaces of the plate. Was this accounted for in the analysis?

A5: Yes, the colonies on the perimeter were excluded from the analysis. We have now clarified this adding a sentence in the material and method section (p16, lines 494-495).

Reviewer 3 (R3)

Q1. One major aspect of this work is that the investigators generated deletion mutants for 92 of the lncRNAs and measured the growth of each mutant under several conditions. However, the authors do not go far enough to show that the observed phenotypes are the result of deleting a lncRNA. The best way to do this would be to generate a complemented strain where the lncRNA is expressed from another locus, as is done when analyzed protein-coding genes. I acknowledge that this may not always work and would depend on the exact mechanism by which the lncRNA is carrying out its function (which at this point is unknown). However, the investigators would at least need to demonstrate that deletion of a lncRNA does not alter the expression of the flanking neighboring genes. This is especially important

given the authors observation that LncRNAs are co-localized with genes involved in azole-sensitivity. Systematically performing qRT-PCR on each of the flanking genes in each mutant, under the appropriate conditions dictated by the phenotype, would go along way towards addressing this concern. For example, if the expression levels of AFUB_031830 and AFUB_031840 were the same when comparing the Wt strain and the AFUBInc_2188.7 deletion mutants grown in the presence of ITR and VOR, one can be convinced that the increased growth is not simply the result of altering the expression of one of this protein-coding genes. The authors begin to address this by citing unpublished results obtained with deletion of AFUB_031830, but fall short. The finding that expression of AFUB_031840 is not altered by azoles is not proof that it is not involved. As currently presented, the assertion that “lncRNAs have functional roles in resistance to azole drugs” is not substantiated by the data.

A1. We considered the approach of expressing the lncRNA from another locus, but we note that most lncRNA effects rely on the positional expression of the lncRNA as build-up of such RNA locally to the expression site plays an important structural role in chromatin 3-dimensional structure and orchestrated expression of chromosomal regions (*Nat Rev Mol Cell Biol* **24**, 430–447, 2023). Hence ectopic expression of lncRNAs is likely to be invalid at best and potentially misleading in the worst case, as also the **R3** acknowledges. *Aspergillus fumigatus* does not support stable plasmid-based expression, preventing plasmid complementation (although some unstable replicative constructs are available). So, we share the **R3**'s view that validation of lncRNA function via checking the phenotype and the expression of the neighbouring genes is the best strategy.

We have checked the effects of the lncRNA deletion on the expression of the flanking genes using qRT-PCR and we have also investigated the the growth rate on solid media of strains carrying mutation on the flanking genes. Given we now present these new data, we removed the reference to the unpublished data.

The experiments have been carried out in two biological replicates (*i.e.* two distinct lncRNA knock-out transformants). In the lncRNA knockout strain, we did not observe differences in expression of both flanking genes during normal growth or in the presence of itraconazole or posaconazole (see Extended Data Figure 6A). In voriconazole the downstream AFUB_013840 is also unaffected in the lncRNA mutant strain, while the upstream gene AFUB_013830 show a ~1 log₂ fold reduction in expression relative to the parental strain. However, when either upstream or downstream genes are knocked out there is no decrease in sensitivity to voriconazole (Extended Data Figure 6B, please see the paragraph below for experimental details). As complete loss of gene function for either upstream or downstream flanking genes does not improve growth in the presence of azoles, we argue that the reduction in expression observed for AFUB_013830 in voriconazole cannot impact the reported azole resistance phenotype that we saw upon deletion of AFUBInc_2188.7.

It is possible, as also the **R3** pointed out, that the lncRNA spans regulatory elements or regions that may impact response of this gene to voriconazole but not the other azoles tested. The reduction in expression is similar to other instances of lncRNA phenotypes observed in other organisms reflecting the difficulty in manipulating any genomic region without affecting neighbouring areas; this issue should also now include consideration of direct effects on lncRNAs when deleting or altering neighbouring genes.

We also performed detailed radial growth measurements on solid media of the flanking gene knockout strains in presence/absence of azoles. If any of neighbouring genes were involved in azole resistance, we would expect to see an increase in radial growth rate in the presence of azole drugs, this is not the case for either neighbouring gene. Specifically, *Aspergillus fumigatus* strains $\Delta AFUB_031830$ and $\Delta AFUB_031840$ (*i.e.* mutants of the neighbouring genes of *AFUBInc_2188.7*), belonging to the COFUN collection of the Manchester Fungal Infection Group, were tested for radial growth measurements in the presence and absence of azoles at the same concentrations used for the original screening of lncRNA mutants (voriconazole, 0.25 mg/L; itraconazole, 0.0625 mg/L; posaconazole; 0.025 mg/L). We did not observe any increase in fitness when comparing both mutants to the A1160 parental control strain, suggesting that phenotype is due to the deletion of the lncRNA alone. While the $\Delta AFUB_031830$ (upstream) displayed similar growth to the wildtype, the $\Delta AFUB_031840$ (downstream), a transcription factor, showed a small but significant growth defect on solid media either with or without azole drugs (Extended Data Figure 6B). In conclusion, there was no improvement in the phenotype when knocking out either of the neighbouring genes.

We have now revised the manuscript to reflect these new set of data. We added a new Extended Figure 6 with the expression and the radial growth data, and added the following text to the manuscript (p9, lines 248-267):

“As the deletion of lncRNA may influence nearby genes, we compared the expression levels of the *AFUBInc_2188.7* flanking genes, namely *AFUB_031830* and *AFUB_031840*, in the WT and the $\Delta AFUBInc_2188.7$ strains. This was carried out in the presence of the three tested azoles (*i.e.* ITR, POS, and VOR) at the concentrations used in the high-throughput screening (0.0625mg/L of ITR, 0.025mg/L of POS, and 0.25mg/L of VOR). The *AFUBInc_2188.7* knockout showed no change in expression for either flanking gene under challenge with ITR or POS and no change in expression for *AFUB_038140* in the presence of VOR (Extended Data Figure 6A). The *AFUB_038130* gene showed a ~1 log₂ fold reduction in expression in the presence of VOR compared to the parental isolate however knockout of either flanking gene did not increase azole resistance or rate of growth on solid media in the presence of any azole (Extended Data Figure 6B). These results are also supported by RNAseq data on WT in the response to itraconazole. The downstream flanking gene (*AFUB_031840*) is not significantly differentially expressed in response to any concentrations of itraconazole treatment, while the upstream flanking gene (*AFUB_031830*) is significantly downregulated in response to itraconazole only when exposed to a very high concentration (4X MIC; LFC -1.8, adjusted p value < 0.0001, Supplementary Table 3). It therefore seems unlikely that deletion of *AFUBInc_2188.7* leads to the azole resistance phenotype via altering expression of the flanking genes but rather through its function as a lncRNA.”